# Prediction of Carbonate Selectivity of PVC-Plasticized Sensor Membranes with Newly Synthesized Ionophores through QSPR Modeling

**Nadezhda Vladimirova** [1], **Valery Polukeev** [2], **Julia Ashina** [1], **Vasily Babain** [3], **Andrey Legin** [1,4]
and **Dmitry Kirsanov** [1,4,*]

1    Institute of Chemistry, Saint-Petersburg State University, Peterhof, Universitetsky Prospect 26, 198504 Saint-Petersburg, Russia; st040543@student.spbu.ru (N.V.); y.ashina@spbu.ru (J.A.); a.legin@spbu.ru (A.L.)
2    JSVC Vekton, Ul. 2-oy Luch 9, 192019 Saint-Petersburg, Russia; polukeev.valeriy@mail.ru
3    ThreeArc Mining LLC, Pr. Narodnogo Opolchenia 10, 198216 Saint-Petersburg, Russia; v.babain@spbu.ru
4    Laboratory of Artificial Sensory Systems, ITMO University, Kronverksky Prospect 49, 197101 Saint-Petersburg, Russia
*    Correspondence: d.kirsanov@gmail.com

**Abstract:** Developing a potentiometric sensor with required target properties is a challenging task. This work explores the potential of quantitative structure-property relationship (QSPR) modeling in the prediction of potentiometric selectivity for plasticized polymeric membrane sensors based on newly synthesized ligands. As a case study, we have addressed sensors with selectivity towards carbonate—an important topic for environmental and biomedical studies. Using the $\log K_{sel}(HCO_3^-/Cl^-)$ selectivity data on 40 ionophores available in literature and their substructural molecular fragments as descriptors, we have constructed a QSPR model, which has demonstrated reasonable precision in predicting selectivities for newly synthesized ligands sharing similar molecular fragments with those employed for modeling.

**Keywords:** ion-selective electrode; selectivity; carbonate sensing; QSPR

## 1. Introduction

Ionophore-based ion-selective sensors are widely employed for quantification of ionic composition in a variety of samples [1,2]. The key element of such sensors is a plasticized polymeric membrane with incorporated ionophore—a lipophilic ligand providing the selectivity towards the target analyte [3]. Currently, the number of ionophores proposed for determination of various inorganic ions is rather large and growing. The search for new ligands is dictated by the needs of their practical application where different types of samples may require quite different selectivity patterns of sensors.

It is noteworthy that the number of ionophores proposed for cations is much higher than that for anions. This relates to the difficulty of developing selective ligands for anion binding: inorganic anions, as compared to cations, are characterized by lower charge-to-radius ratios and a large variety of geometries, and the forms of anion existence in the solution strongly depend on pH [4]. The sensor membrane phase formed by the plasticized polymer is very hydrophobic, so it is difficult for hydrophilic ions to cross the phase boundary, and thus, ensuring the selectivity of polymer sensors that differ from the Hoffmeister series is a major challenge.

The common experimental protocol for studying novel potential ligands for ion-selective sensors requires chemical synthesis of candidate substances, their purification and characterization, sensor membrane preparation (typically in a broad variety of compositions including different solvent-plasticizers and different ratios between a ligand and an ion-exchanger), and potentiometric measurements to assess the sensitivity and

selectivity parameters of the novel membranes. The overall process is quite tedious and time-consuming and the success is not guaranteed, i.e., the studied candidate substances may have no required selectivity for further use.

In these circumstances, it would be good to have an instrument that would allow an initial screening of candidate ionophore substances and, based on the chemical structure of the substance, would be able to reliably predict if the particular ligand is worth the whole experimental study. This in silico approach would save a considerable amount of time and resources in developing new sensors.

Quantitative Structure-Property Relationship (QSPR) is a computational method that aims to build a mathematical model relating the chemical structure of the substance to some of its properties [5]. In order to do so, the QSPR approach formalizes a chemical structure in a set of molecular descriptors (numbers that describe the structure as a whole [6]) and tries to find a mathematical relation (regression model), connecting these descriptors with a property of interest. The model construction requires a training set of molecules where the target property under study is experimentally determined. QSPR modelling is a mature field of research that has contributed to many different areas of chemistry such as, e.g., drug design [7,8], material science [9,10], and toxicity evaluation [11,12]. QSPR can effectively consider various complex effects in multicomponent systems. For example, QSPR was successfully applied for prediction of equilibrium constants in the complexation of metals with organic ligands and in liquid extraction processes [13,14].

Recently, the QSPR approach was suggested for the prediction of analytical performance parameters for ionophore-based potentiometric sensors [15,16]. The potentiometric sensitivity of the sensors based on a variety of nitrogen-containing ligands (mainly diamides of pyridine and bipyridine acids) towards heavy metal ions ($Cu^{2+}$, $Zn^{2+}$, $Cd^{2+}$, $Pb^{2+}$) was predicted with root mean squared errors around 5 mV/dec based on the set of descriptors derived from substructural molecular fragments [15]. This model was based only on the experimental data obtained by the authors. The study [16] developed this concept further and aimed at $Mg^{2+}$/$Ca^{2+}$ potentiometric selectivity prediction using QSPR based on the literature data. While the attained precision of the model in prediction was not very high ($\pm 0.5$ logK$^{sel}$), the model was able to distinguish reliably between the ligands with high, medium, and low $Mg^{2+}$ selectivity.

The present work is devoted to study whether it will be possible to predict potentiometric selectivity of the sensors based on newly synthesized ligands using a QSPR model based on literature data. As a challenging case for modeling, we have focused on carbonate-selective sensors, where the number of available ligands with appropriate selectivity is rather limited. The selectivity values logK$^{sel}$($HCO_3^-$/$Cl^-$) were used as a target parameter for modeling.

## 2. Materials and Methods

### 2.1. The Dataset for Modelling

The following flow diagram (Figure 1) illustrates the logic of the experiment.

The dataset for QSPR modeling contained the carbonate ionophores described in the literature (40 substances in total). The structures of all ionophores obtained from the literature sources are given in Table S1 (Supplementary Materials). Considering the small number of papers reporting on the successful development of carbonate potentiometric sensors and in order to extend the dataset with additional entries, we have also included several ionophores showing poor carbonate selectivity. In these cases, the reported values logK$^{sel}$($Cl^-$/$HCO_3^-$) were converted into logK$^{sel}$($HCO_3^-$/$Cl^-$) according to Nikolsky-Eisenman equation:

$$E = E_I^0 + \frac{RT}{z_I F}\ln(a_I + \sum K_{IJ} a_J^{\frac{z_I}{z_J}}),$$　　　(1)

where $a_I$, $a_J$, $z_I$, $z_J$ are the activities and charges of the primary and interfering ions, respectively, and $K_{IJ}$ is a selectivity coefficient. The value of the selectivity coefficient can be found according to:

$$K_{IJ} = a_I / (a_J)^{Z_I/Z_J}, \qquad (2)$$

and it is possible to calculate the logarithm of the selectivity coefficient $K_{JI}$ given that the charges of the ions I and J are equal. Thus, in our case, $\log K^{sel}(Cl^-/HCO_3^-) = -\log K^{sel}(HCO_3^-/Cl^-)$.

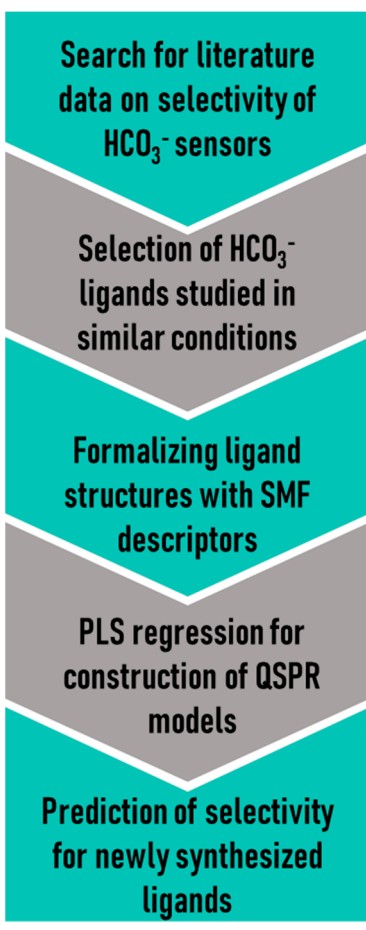

**Figure 1.** The sequence of the experiment.

While compiling the literature dataset we have limited ourselves with the papers where poly(vinylchloride) (PVC) was employed as a polymer for membrane matrix. Since the solvent-plasticizers varied from study to study, we have only considered the papers employing *o*-nitrophenyl octyl ether (NPOE), dioctyl adipate (DOA), or bis(2-ethylhexyl) sebacate (DOS) as plasticizers. In order to take into account the differences in membrane properties induced by the plasticizer, its dielectric permittivity was added to the dataset as one of the descriptors for modelling. The anion-exchanger employed in all the literature sources was tridodecylmethylammonium (TDMA). In order to ensure that $HCO_3^-$ is a dominant ionic form in solutions, we only considered the literature data obtained in the pH range 7.0–8.6. All $\log K^{sel}(HCO_3^-/Cl^-)$ were obtained using a separate solution method, except for eight substances from [17] where the matched potential method was employed. Since these two methods usually yield comparable results, it was decided to keep these eight entries in the dataset. The uniformity of experimental conditions is very important to warrant a reliable dataset for QSPR modelling. The selectivities $\log K^{sel}(HCO_3^-/Cl^-)$ in the literature dataset varied from −5.8 to 6.2. The mean value was −1.4 and the median value was −2.6.

The structures of the new ligands are given in Table 1. The motivation for synthesis of these particular substances was the following: most of the carbonate ionophores developed so far are based on trifluoroacetophenone and its derivatives. The working fragment responsible for carbonate ion binding is the carbonyl group with lowered electron density on oxygen due to the presence of a strong acceptor—the CF$_3$ group. The substance **1** is commercially available and was acquired from Merck (Darmstadt, Germany) and used as received. The ligands **2–4** (Table 1) were newly synthesized. All four substances contain different acceptor substituents in the vicinity of carbonyl. We have hypothesized that these ligands may also have considerable carbonate binding ability.

The synthesis and characterization of newly synthesized substances are provided in the Supplementary Materials along with their NMR spectra.

**Table 1.** The new ligands employed for the study.

| # | Ligand Structure | IUPAC Name |
|---|---|---|
| 1 |  | Phenanthrenequinone **S1** |
| 2 |  | 1-(diphenylmethyl)-1H-indole-2,3-dione **S2** |
| 3 |  | 2-((4-chlorophenyl)(phenyl)methyl)-2-nitro-1H-indane-1,3-dione **S3** |

**Table 1.** *Cont.*

| # | Ligand Structure | IUPAC Name |
|---|---|---|
| 4 |  | 5,7-dimethoxy-2-benzyl-1,3,4(2H)-isoquinolinetrione **S4** |

### 2.2. Potentiometric Sensors Based on New Ligands

Four new ligands were employed for preparation of PVC-plasticized sensor membranes. The compositions of the membranes are given in Table 2. Each membrane contained 50 mmol/kg of ionophore and 10 mmol/kg of TDMA-NO$_3$ as the anion-exchanger. The ratio between PVC and a solvent-plasticizer was 1:2 and the total weight of membrane was 300 mg. *o*-nitrophenyloctyl ether (NPOE) was employed as the solvent-plasticizers.

**Table 2.** Sensor membrane compositions (wt%).

| Sensor | PVC | TDMA NO$_3$ | Plasticizer | Ligand |
|---|---|---|---|---|
| S1 | 32.79 | 0.60 | 65.57 | 1.04 |
| S2 | 32.61 | 0.60 | 65.22 | 1.57 |
| S3 | 32.48 | 0.60 | 64.96 | 1.96 |
| S4 | 32.59 | 0.60 | 65.18 | 1.63 |

Polymeric sensor membranes were prepared according to the standard protocol. The weighted amounts of all membrane components were dissolved in 3 mL freshly distilled tetrahydrofuran (THF) in a glass beaker using a magnetic stirrer. The resulting solutions were poured into teflon beakers 20 mm in diameter and left to dry for 48 h. The disks 8 mm in diameter were cut from the parent membranes and then attached to the PVC sensor bodies with a mixture of PVC and cyclohexanone. The thickness of the prepared membranes was 0.4 mm. After drying the glue, the inner parts of the resulted electrodes were filled with a mixture of 0.01 M NaHCO$_3$ and 0.001 M NaCl. The chloride anion is required for Ag/AgCl electrode functioning, while bicarbonate is needed to ensure its constant content in the membrane phase required by Nikolsky formalism. The water for making all aqueous solutions was obtained from distiller GFL 2102 (GFL Burgwedel, Germany). The conductivity of the mono-distillate is 2.2 µs/cm at 25 °C. Finally, internal Ag/AgCl electrodes were embedded in the sensors. Three replicate sensors were prepared from each membrane composition.

Potentiometric measurements were performed with a multichannel digital mV-meter KHAN-32 (Sensor Systems LLC, St. Petersburg, Russia) connected to a PC for data acquisition through USB port. The measurements were done against the standard reference Ag/AgCl electrode ESr-10101 (Izmeritel'naya Tekhnika, Moscow, Russia). The glass pH sensor PY-41 (GOnDO Electronic Co., Ltd., Taipei, Taiwan) was employed to control pH values in sample solutions.

Sensor sensitivities were studied in aqueous solutions of inorganic salts (sodium sulphate, sodium chloride, sodium nitrate, sodium bicarbonate, and monosodium phosphate) in the concentration range from $10^{-6}$ to $10^{-2}$ M. The sensitivity values were calculated as the slopes of the linear parts of sensor response curves ($10^{-4}$–$10^{-2}$ M). The sensors were

washed with several portions of distilled water before, after, and between the measurements until the constant values of potential was achieved.

Selectivity coefficients were obtained by separate-solution method, which is also known as bi-ionic potentials method [18]:

$$\log K_{IJ} = \frac{z_I F(E_J - E_I)}{RT \ln 10} + (1 - z_I/z_J) \log a a_I. \tag{3}$$

$E_I$ and $E_J$ were registered in $10^{-3}$ M solutions of primary and interfering ions, respectively.

### 2.3. QSPR Modelling

Substructural Molecular Fragments (SMF) were employed as descriptors for encoding molecular structures of ligands. This method interprets a molecular structure as a graph so that all possible molecular fragments are subgraphs. These fragments with given length are found and counted and their numbers are placed in a resulted descriptor matrix.

"MolFrag", which is a part of "ISIDA QSPR" software suite was used to obtain SMF [19]. SMF in ISIDA can be described in two ways: either as sequences of atoms and/or bonds (topological path) or as a selected ("augmented") atom (atom-centred fragments) with its environment, which can be atoms, bonds, or both of them. Both ways are illustrated in Figure 2.

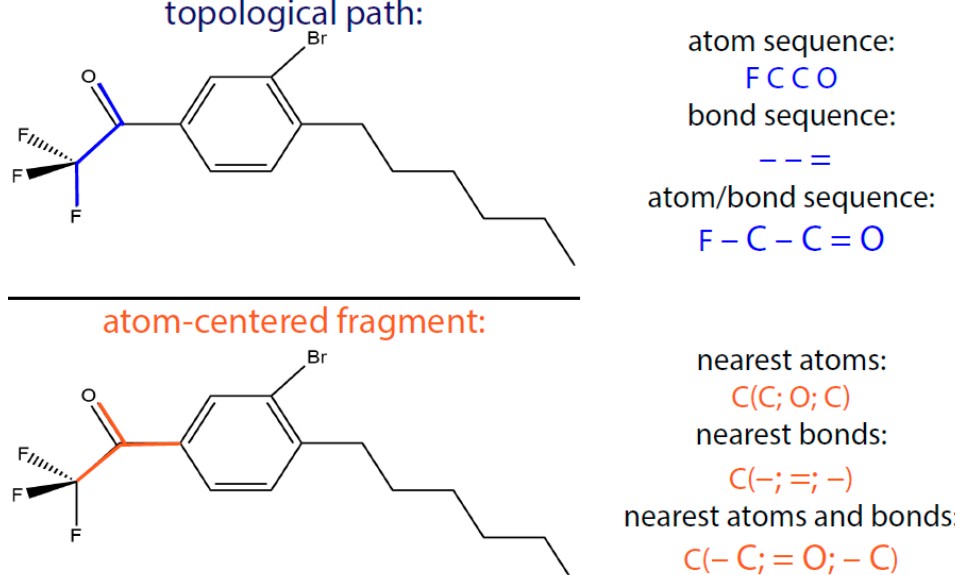

**Figure 2.** Two approaches for obtaining substructural molecular fragments.

In this research we chose atom and bond sequences for molecular description. Only the shortest paths from one atom to the other were considered. The minimal and the maximal lengths of atom-bonds chains in this research were 4 and 10, respectively.

The generation of descriptors resulted in the matrix 40 [number of literature compounds] × 1103 [calculated structural descriptors], which was extended with one more column containing dielectric constants of the solvent-plasticizers employed in the sensor membrane preparation. The resulting matrix was used to establish the regression model relating these descriptors with $\log K^{sel}(HCO_3{}^- /Cl^-)$ selectivity values of the corresponding potentiometric sensors.

Partial Least Squares (PLS) algorithm was applied for constructing multivariate regressions. PLS is a very popular chemometric tool for such applications and it is widely applied in QSPR studies [20–22]. Briefly, PLS seeks to calculate the coefficients ***B*** in the equation:

$$\boldsymbol{Y = BX,} \tag{4}$$

where $Y$ is a vector containing target property values ($logK^{sel}(HCO_3^-/Cl^-)$ values in this study) for each of the cases in the training set and $X$ is the matrix containing descriptive variables for each of the cases (molecular descriptors in this study). The PLS algorithm is looking for the variables in descriptors that would be highly correlated to the variance in the $Y$ vector of the training data. These correlated variables will get the highest values of regression coefficients. The detailed description of PLS can be found elsewhere [23].

PLS models were calculated in The Unscrambler 9.7 software package (CAMO, Norway). Full cross-validation was applied to optimize the number of latent variables and the number of descriptors in the model using RMSECV value (root mean square error of cross-validation) as a criterion:

$$RMSEC = sqrt\ ((sum(y_{pred} - y_{real})^2)/n), \tag{5}$$

where $y_{pred}$ and $y_{real}$ are the predicted and real values of the target property in each run of full cross-validation and $n$ is the number of samples.

Prior to modelling, the values of descriptors in the $X$ matrix were autoscaled (the column wise average was subtracted from each element and the result was divided by column wise standard deviation).

The resulted model was used to predict the $logK^{sel}(HCO_3^-/Cl^-)$ values for the newly synthesized ligands.

## 3. Results and Discussion

The matrix of molecular descriptors acquired for 40 ligands was related to their $logK^{sel}(HCO_3^-/Cl^-)$ values using PLS regression. The initially obtained model was optimized with respect to the number of variables using regression coefficients values in the model. All variables having regression coefficients in the range $[-5 \times 10^{-3}; 5 \times 10^{-3}]$ were excluded from consideration, thus yielding 585 variables in the model. The resulting QSPR model is given in Figure 3. Based on the model statistics (RMSECV and $R^2$ values) it can be concluded that the derived relation is suitable for semi-quantitative prediction of potentiometric selectivity of ionophores. The $logK^{sel}(HCO_3^-/Cl^-)$ values varied from −6 to +6 in the modeled dataset and RMSECV value is 1.5, thus, the model allows distinction between poorly, medium, and highly selective carbonate ligands. This appears to be a promising result considering a wide chemical diversity of the ligands, comparatively small (in QSPR scale) dataset, employment of literature data obtained in obviously non-identical conditions, and overall simplicity of the approach.

The analysis of the regression coefficients of the PLS model allows judging on the importance of particular descriptors and their contribution to the selectivity values. The largest contribution is made by the fragments having the highest absolute values of regression coefficients. Calculated fragments (model variables) were filtered by the presence of at least five ligand structures. Then, the remaining fragments were analysed for repeatability in other fragments, and the smallest fragments were chosen from the row of equivalent fragments. In this way, the fragment C-C-C-O-C=C contains the shorter fragment C-O-C=C, and this shorter fragment was kept for analysis. Finally, the fragments were sorted by the value of the corresponding regression coefficients and only the fragments with absolute values of b < 0.01 were considered. The resulting variable importance graph is shown in Figure 4.

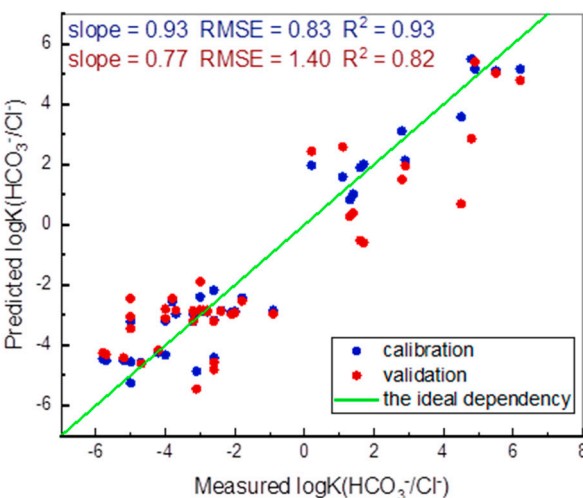

**Figure 3.** "Measured vs predicted" plot of cross-validated QSPR model for predicting the selectivity of membrane sensors. The green line corresponds to the ideal match between measured and predicted values.

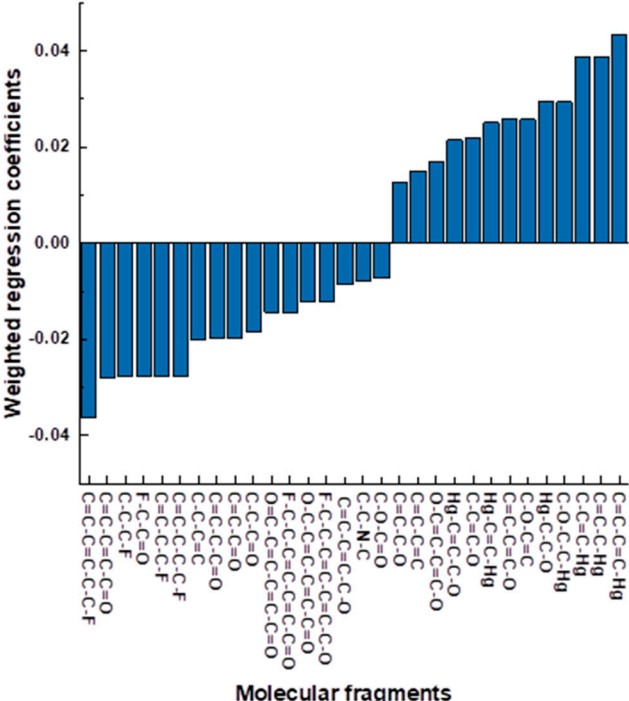

**Figure 4.** Fragments with the largest contribution in $logK^{sel}(HCO_3^-/Cl^-)$ values, where '-' is a single bond and '=' is a double bond.

It can be seen that the largest negative contribution corresponds to the fragment C=C-C=C-C-C-F. Since the modelled value was $logK^{sel}(HCO_3^-/Cl^-)$, and the more negative it is the higher is the selectivity towards $HCO_3^-$, this means that such C=C-C=C-C-C-F fragments contribute to greater selectivity towards the hydrocarbonate anion. It should be noted that amongst the 17 fragments with the negative contribution, there are only 2 that have no fluorine ($F^-$) or C=O groups. These fragments are C=C-C=C-C-O and C-C-N-C and they also have significant negative contributions amongst the chosen fragments (15th and 16th place out of 17).

Ten fragments with the largest negative contributions occur in the trifluoroacetophenone (TFA) group. The chemical structure of this group is shown in Figure 5. The TFA group is part of a significant amount of existing carbonate ionophores. The great exam-

ple of ionophore with this group and the great selectivity ($logK^{sel}(HCO_3^-/Cl^-) = -3.1$) is N, N-Dioctyl-3$\alpha$, 12$\alpha$-bis(4-trifluoroacetylbenzoyloxy)-5$\beta$-cholan-24-amide (carbonate ionophore VII from Merck catalogue [24]). The structure of this compound can be found in Table S1 in the Supplementary Materials. The TFA group is known to promote carbonate anion bonding through the formation of hydrogen bonds [25].

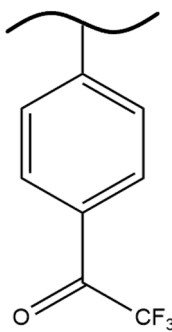

**Figure 5.** The chemical structure of the trifluoroacetophenone group.

The fragment with the greatest positive contribution is C=C-C=C-Hg. Only 6 fragments amongst the 13 fragments with the positive contribution have no mercury (Hg$^-$) in their compositions. In descending order of positive contribution, these fragments are C-O-C=C, C=C-C=C-O, C-C=C-O, O-C=C-C=C-O, C=C-C=C, and C=C-C-O. These fragments come from the ionophores proposed for chloride sensing, and thus have low selectivities towards carbonate [26]. An example of these ionophores is {µ-[4,5-Dimethyl-3,6-bis(octyloxy)-1,2-phenylene]} bis(trifluoroacetato-0) dimercury, which has all the fragments with positive contribution in its structure, which results in $logK^{sel}(HCO_3^-/Cl^-) = 5.5$. The structure of this ligand can be found in Table S1 (Supplementary Materials).

Thus, it can be stated that the constructed QSPR model agrees well with the chemical considerations on ligand structures and the importance of various fragments corresponds to the chemical intuition.

The polymeric sensor membranes prepared with four new ligands were studied with respect to their sensitivity towards inorganic anions and their $HCO_3^-/Cl^-$ selectivity. The typical response curves of the new sensors are given in Figure S1 (Supplementary Materials) and the calculated sensitivity values are shown in Table S2 (Supplementary Materials). It must be pointed out that the **S2** and **S3** sensors did not demonstrate sufficient sensitivity towards carbonate.

Using the QSPR model derived from literature data, the selectivities of these four new sensor membrane compositions were predicted. In order to do so, each of the ligands were described with the same set of SMF and the resulting row-vectors were filtered to put them into the correspondence with the filtered set of 585 variables. The resulting data were plugged into the QSPR model, which returned the predicted $logK^{sel}(HCO_3^-/Cl^-)$ values. The results of this prediction are given in Table 3 along with the data on selectivity that were obtained in a traditional potentiometric experiment using a separate solution method.

**Table 3.** The comparison of the predicted and the experimental $logK^{sel}(HCO_3^-/Cl^-)$ values ($\pm0.1$ logK for experimental data) for four newly synthesized ionophores.

| Sensor | $logK^{sel}(HCO_3^-/Cl^-)$ Experimental | $logK^{sel}(HCO_3^-/Cl^-)$ Predicted |
|--------|------------------------------------------|---------------------------------------|
| **S1** | −3.6 | −2.5 |
| **S2** | −0.7 | −3.4 |
| **S3** | −1 | −2.9 |
| **S4** | −2.5 | −2.7 |

Taking into account the RMSECV value of the QSPR model of 1.4 $logK^{sel}(HCO_3^-/Cl^-)$, the correspondence between the predicted and the experimental selectivities appears to

be good in three out of five cases (**S1**, **S4**, **S5**). The **S1** compound has C=C-C=C-C=O, C-C-C=C, C=C-C-C=O, and C=C-C=O, C-C-C=O fragments with negative contribution and a C=C-C=C fragment with a positive contribution. The model predicted the logarithm of the selectivity coefficient with a value of $-2.5$ while the experimentally found selectivity coefficient is $-3.6$. The **S2** compound contains a C-C-N-C fragment and the rest of the fragments are the same fragments with the first compound. All of these fragments, except C=C-C=C, make a negative contribution to the logarithm of the selectivity coefficient. Thus, there is a noticeable deviation between the predicted $\log K^{sel}(HCO_3{}^-/Cl^-) - 3.4$ and the experimental value of $-0.7$. The **S3** compound has only a few fragments with a significant model impact. These fragments are C-C-C=C with the negative contribution and C=C-C=C with a positive contribution. As the model has only two structural fragments with a significant impact for this compound, the difference between the prediction of the model which $(-3.0)$ and the experimental data $(-1.0)$ is also high. A rather high discrepancy between prediction and experiment for **S2** and **S3** can also be explained by the fact that these sensors in fact did not have reasonable response towards carbonate (Table S2).

The **S4** compound has fragments such as C=C-C=C-C=O, C=C-C-C=O, C=C-C=O, C-C-C=O, C-C-C=C, O-C-C=C-C=C-C=O, and C-C-N-C with a negative contribution and C-O-C=C, C=C-C=C-O, C-C=C-O, and C=C-C=C with a positive contribution in its composition. Since the number of important fragments is rather high, the agreement between the experiment and the calculation is also rather good $(-2.7$ vs. $-2.5$, correspondingly).

Thus, proper prediction requires that the ligands employed for QSPR model construction would share the same important structural fragments with newly proposed ligands. This issue is somehow obvious from the chemical and mathematical intuition as the diversity of the training set determines the applicability domain of the model.

## 4. Conclusions

The QSPR model relating the structure of carbonate ionophores to the selectivity $(\log K^{sel}(HCO_3{}^-/Cl^-))$ of the corresponding potentiometric sensors with plasticized polymeric membrane was developed based on the literature data on 40 ligands. The important structural fragments in ligands were identified using the regression coefficients values in the PLS model. The applicability of the model for prediction of the selectivity of four new ligands with different acceptor substituents at carbonyl group was explored. For two ligands that have demonstrated potentiometric sensitivity to hydrocarbonate ions, reasonable agreement in predicted and experimental selectivities was observed. It is noteworthy that the match with experimental data was better for the ligands sharing the same important molecular fragments with those employed for QSPR model construction. We believe these results demonstrate the potential of QSPR approach in the development of novel anion-selective sensors.

**Supplementary Materials:** The following supporting information can be downloaded at: https: //www.mdpi.com/article/10.3390/chemosensors10020043/s1, Figure S1: Typical potentiometric response curves of the sensor S2; Table S1: Structure and selectivity of the ionophores employed for modeling; Table S2: Sensitivity values of the sensors in the solution of the studied anions, mV/dec.

**Author Contributions:** Conceptualization, D.K., V.B.; methodology, D.K.; formal analysis, D.K., N.V.; investigation, N.V., V.P.; resources, A.L., V.P.; writing—original draft preparation, N.V., V.P.; writing—review and editing, D.K., V.B. and A.L.; visualization, N.V.; supervision, D.K.; funding acquisition, J.A. All authors have read and agreed to the published version of the manuscript.

**Funding:** This research was supported by RFBR project #20-33-70272.

**Conflicts of Interest:** The authors declare no conflict of interest. The funders had no role in the design of the study; in the collection, analyses, or interpretation of data; in the writing of the manuscript, or in the decision to publish the results.

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
