# Peer review of "Prediction of Carbonate Selectivity of PVC-Plasticized Sensor Membranes with Newly Synthesized Ionophores through QSPR Modeling"

_chemosensors, doi:10.3390/chemosensors10020043_

Round 1

Reviewer 1 Report

In my opinion, manuscript entitled: „Prediction of carbonate selectivity of PVC-plasticized sensor membranes with newly synthesized ionophores through QSPR modeling” presents an interesting approach to search and apply a new ionophors for ion-selective electrodes. Research presented by the authors are good example of usage mathematical methods for design of potentiometric sensors. The wide spectrum of the analyzed ionophores deserves to be emphasized. However, this work requires some improvements. Below there are comments and questions for the authors.

  1. The wrong formula is used to write the selectivity coefficient. The formatting of superscripts and subscripts is incorrect.
  2. What is “bionic potential method”? Is it correct name?
  3. The authors wrote in their manuscript that "The synthesis and characterization of newly synthesized substances are provided in Supplementary Material along with their NMR spectra”. I found only a synthesis method in the description. However, there is no description of NMR spectra confirming the correctness of the obtained structures.
  4. What was the thickness of the prepared ion-selective membranes?
  5. Why did it take 48 hours to dry the membranes?THF is a very volatile solvent and 24 hours are sufficient for membrane preparation.
  6. Why authors decided to use a mixture of PVC and cyclohexanone?
  7. How was the composition of the internal solution selected for ISEs construction?
  8. In the Fig. S1. presented in the Supplementary Materials, the authors did not provide the units for the x-axis.
  9. Research with the use of ISEs, and especially their metrological characterization, requires the use of very pure water.The authors wrote that it was distilled water. Please state how the water was obtained and what parameters it had (purity class or electrical conductivity).
  10. Formula (3) presents a simplified method of calculating the selectivity coefficient (SSM - separate solution method). The text contains the name of the bionic potential method. This introduces a certain inaccuracy that makes it difficult to receive work. Moreover, the FIM method is the currently recommended method for determining the selectivity coefficients.The SSM method works well when the characteristics of the sensor in relation to interfering ions show a Nernst response.
  11. There is an incorrect regression coefficient at the beginning of a paragraph in Chapter 3
  12. Table 3 shows the experimental results - there are no standard deviations. How many measurements and how many electrodes were used to determine the selectivity coefficients?
  13. In Table 3, the authors should add a column containing information about the sensitivity of the electrodes.
  14. A flow diagram would make it easier to understand the algorithm used for the modeling.
  15. In table S2, sensitivity should be recorded in accordance with IUPAC standards (one-tenth approximation)

Other notes relate to minor fixes such as punctuation marks and excess spaces.

Reviewer 2 Report

The authors made a huge work. I find it very interesting. I have few comments:

  • why the authors did not give a slope of prepared membranes and belonging concentration ranges?
  • did the authors find a lifetime of each prepared membrane according the stability of an ionophore?
  • I did not find methods about purification of synthesised compounds from reaction mixtures in supplement materials. This is very important
  • line 300 (table's caption): should it be written four instead five?
  • there are only two reference two years or less young. that is about 7 % of all references. this can be considered as indicator for low interest for this topic. I find this topic very interest and the authors must use more recent references.

Round 2

Reviewer 1 Report

Thank you for the answers that I accept. I have a general comment to the authors that should be considered before the next potentiometric research.  ISEs parameters determination requires the use of high purity water. I recommends the use of reverse osmosis water, the purity of which is approximately one hundred times higher than that of distilled water. Ions contained in distilled water may affect the metrological parameters of the sensors, and higher measurement errors may be generated.

Reviewer 2 Report

The authors obeyed the reviewer's comments and built them in a manuscript or gave proper comments. Therefore, I suggest accepting the manuscript and publishing in current form.